# Assessing the Impact of Park Renovations on Cultural Ecosystem Services

**Xin Cheng [1,*]**, **Sylvie Van Damme [2]** and **Pieter Uyttenhove [3]**

1   Department of Urban Planning and Landscape Architecture, Xihua University, 999 Jinzhou Road, Pidu District, Chengdu 610039, China
2   School of Arts, University College Ghent, 9000 Ghent, Belgium; sylvie.vandamme@hogent.be
3   Department of Architecture and Urban Planning, Ghent University, 9000 Ghent, Belgium; pieter.uyttenhove@ugent.be
*   Correspondence: xin.cheng@mail.xhu.edu.cn; Tel.: +86-15908142467

**Abstract:** Urban parks are essential in enhancing the quality of city life by providing diverse cultural ecosystem services (CES). Despite considerable investments in park renovations, there is very little evidence about whether park renovations can properly secure CES. This study provides a basis for the incorporation of CES evaluation into urban park practice to maximize CES. We specifically ask how CES are influenced by park renovations. We developed a participatory mapping approach by asking people to assess CES on a current map and on a historical map, representing the situations before and after the renovation, instead of doing a follow-up study, in order to be more time-efficient and enhance the comparative effects. The results show that the park renovations had different impacts on CES and not all the renovations had positive impacts. This study has a huge potential for supporting park practice. First, this study shows that specific park renovations can be used to guide park management to enhance CES. Second, this study provides a new insight for landscape architects to rethink their design proposals before construction. Third, the study encourages the method of combining participatory mapping and interviews to link CES to a specific location and specific renovations.

**Keywords:** cultural ecosystem services; park renovation; participatory mapping; park design; park management

## 1. Introduction

It is well recognized that urban parks provide a range of cultural ecosystem services (CES), which refer to the non-material benefits that people receive from ecosystems such as recreation and aesthetic value [1]. CES highlight the multiple values that stakeholders attach to ecosystems, especially by eliciting socio-cultural values [2], and they are recognized as playing a significant role in improving human wellbeing and promoting environmental sustainability [3–6]. Urban CES are defined here as those services that are provided by urban ecosystems and their components. CES not only derive from ecological properties and processes, but also from participation and modification by humans [7,8]. This is why the evaluation of CES in urban context, which have a high level of heterogeneity, is more complex than evaluating natural ecosystems that have a relatively homogeneous environment (e.g., forests, marine areas, or farmland) [8,9]. CES are regarded as the most human-made ecosystem services (ES) [10] and their evaluations highly depend on people's perceptions and preferences [11,12]. Voigt and Wurster [13] point out that CES have to focus more on people than on ecosystems. Various studies have been conducted to assess CES in urban parks. For example, Campbell et al. [14] examined recreation, social relations, and sense of place in a park by researching the park's visitors. Although CES in urban parks receive considerable academic attention, the assessments often focus on one or two services (e.g., recreational and aesthetic values), and they often underestimate the importance of

assessing ES [15,16]. Moreover, the exploration of their potential to support park practice has been unsatisfactory because this is difficult to evaluate due to their abstract and intangible characteristics. There is a lack of commonly accepted framework for analyzing CES, characterizing their changes, and integrating them into the ES framework [17].

Park renovation, also known as park refurbishment, park renewal or park improvement, refers to a process that aims to address various issues, such as improving park quality, increasing park usage, and solving environmental problems [18], by modifying park settings and optimizing a park's design. Park renovation has received considerable investment to enhance the quality of city life and the environment because of the great pressure of the rapid urbanization. However, there is very little evidence about whether park renovation can properly secure CES since assessing the impact of renovation on CES has proven to be a complex undertaking [19]. Whether the park renovation contributes positively or negatively to CES is unknown without an evaluation. The existing studies mainly focus on the impacts of park renovation on physical activity or aesthetic values by means of pre- and post-evaluations or landscape performance evaluations [20]. For example, Veitch et al. [21] investigated whether the changes of the physical environment increased overall park usage and park-based physical activity by observing park visitors before and after the park's renovation. Vert et al. [22] conducted systematic observations of park users before and after the intervention in order to quantify and compare the changes of physical activity. However, follow-up measurements are time-consuming [23]. Moreover, other types of CES, in addition to recreation and aesthetic values, are very important and should also receive attention to enhance the assessment accuracy [24]. Hence, this study focuses on assessing how CES are influenced by park renovations through an effective approach which is less time-consuming.

A growing number of studies have been conducted on integrating CES evaluations into land-use planning, urban planning, and urban green infrastructure planning. For example, Kati and Jari [25] assess the CES of blue and green infrastructure in Helsinki, Finland, and evaluate how CES could be integrated into the early stage of green area planning. For more examples, see Andersson et al. [26], Gómez-Baggethun and Barton [27], Jansson [28], Kabisch et al. [29], Mascarenhas et al. [30], and Woodruff and BenDor [31]. However, few efforts have focused on systematically investigating CES in supporting landscape architecture design. This is a problem because landscape architects play a significant role in maintaining ecosystem services and human well-being through their daily designs of urban blue, green, and built-up spaces. Another growing number of studies investigating the relationship between CES and landscape features gives an opportunity to address this gap [5,32,33]. Those studies provide insights into the location of CES supply and their correlations with specific features such as vegetation, benches, or recreation facilities through participatory mapping [34]. This study is inspired by these cases and assumes that specific park renovation has an influence on CES and has the potential to support park practice (design or management). We also study how specific park renovations affect CES change. The concept of "participatory mapping" includes any process where people share in the creation of a map [35]. In the mapping process, participants identify spatially explicit benefits from the given map that contribute to human well-being. Participatory mapping may also include an assessment of the relative importance of the services provided and the change of the services [35]. For instance, Jaligot et al. [36], adopted this method to understand how CES has changed over time, as a response to urbanization in Cameroon.

This study aims to evaluate the impact of park renovation on CES, and we specifically ask how CES are influenced by park renovation. We also ask by what specific renovation they are influenced. In addition, we further discuss how this knowledge can support the practice, such as the park design and management. To achieve this, we further developed the participatory mapping approach by improving time efficiency and enhancing the comparative effects, and we transferred the abstract CES into a more practical application. Specifically, we conducted the comparative study after the renovation by asking people to assess CES on a current map and on a historical map, which showed the situations before

and after the renovation. Then, we further asked participants through interviews to identify what specific park renovations influenced their perceptions. We conducted the case study in an urban park in Chengdu, China.

## 2. Materials and Methods

### 2.1. Study Area

Given the fast economic and societal changes in China in recent decades, cities have faced massive redevelopment, renovation, and restructuring [37]. These changes offer an opportunity for landscape architects to improve environmental quality, to achieve sustainable development, and to meet public needs through the renovation of urban green spaces [38]. This study was undertaken in Huanhuaxi Park (Figure 1), which is the largest urban park (36.92 hm$^2$) in central Chengdu, the administrative and cultural center of Sichuan province, located in southwest China. The Huanhuaxi Park was built in 2003. It is located between the First Ring Road and the Second Ring Road, neighboring Sichuan Museum to the east, and the Thatched Cottage of Du Fu (a famous poet during the Tang dynasty) to the north. This forms a natural link between the top two attractions of the city. The renovation belongs to the Xijiao River Project and the Qianshibei Project, both announced by the Chengdu government, and designed by the Sichuan Provincial Architectural Design and Research Institute. The goals of the project included: improving the landscape quality of Huanhuaxi Park, particularly its water quality, and promoting traditional culture and poetry by highlighting poems written by Du Fu (1455 in total). The renovation lasted from 2016 to April 2018. The park was partly closed during the renovation period, and sections under construction were enclosed by walls and inaccessible to all users.

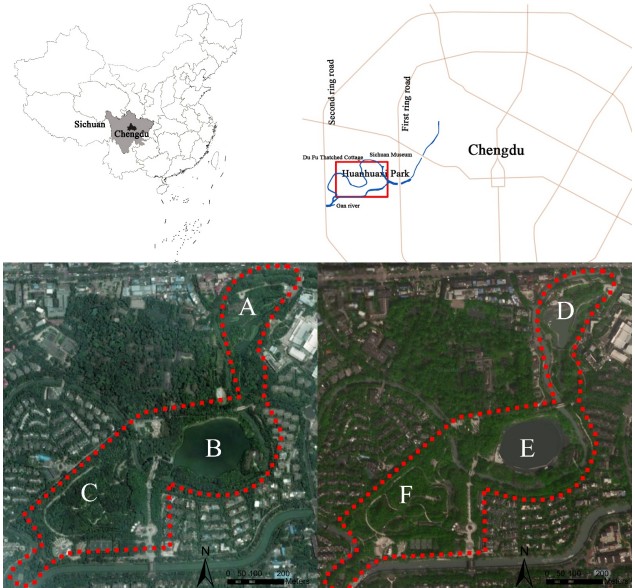

**Figure 1.** The images show a comparison of the park in the year 2015 and 2018 representing the park before and after the renovation: (A) Egret Island in 2015; (B) Canglang Lake in 2015; (C) Wanshu Hill in 2015; (D) Egret Island in 2018; (E) Canglang Lake in 2018; (F) Wanshu Hill in 2018.

Huanhuaxi Park mainly contains a lake (Canglang Lake), a wetland (Egret Island), a hill (Wanshu Hill), and streams meandering across the area. In addition, plenty of scenic spots and facilities are spread throughout the park. Table 1 shows the park features before and after renovations. The basic layout (lake, hill, and wetland) of the park were not changed, while recreational facilities, plants, trails, paths, and other infrastructure were replaced, and new structures were built (e.g., pavilions). Only some large trees and major structures (e.g., the tennis court) were retained. Other updates in the other part of the park included new sculptures, benches, plants, and restoration of the streams.

**Table 1.** Main renovated locations.

| Locations | Before Renovation | After Renovation |
|---|---|---|
| **Canglang Lake** | 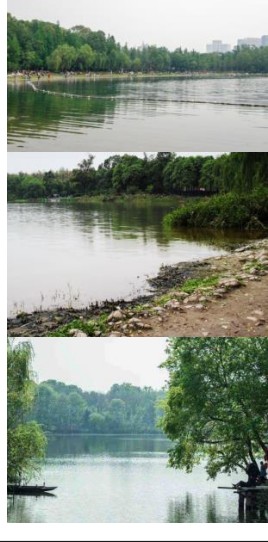 Canglang Lake lies in the center of Huanhuaxi Park. It is composed of a square connected to the Thatched Cottage of Du Fu and an island where Huanhua Hall was built. The natural landscape comprises of willows, rockeries, and statues along the bank. | 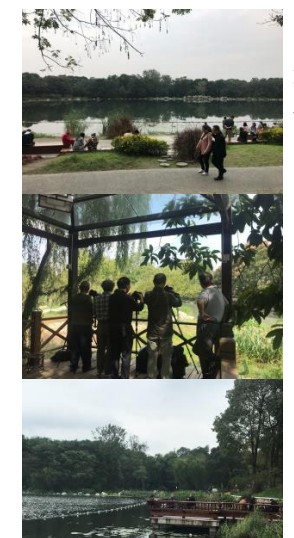 The main part of the renovation project was the significant overhaul of Canglang Lake, including the water body, trail, seating areas, sculptures, and plants. Specifically:<br><br>• The lake was ecologically restored, including purified water, and new aquatic habitat for plants and animals. The restoration process is shown on a bulletin board by the lake.<br>• Different kinds of benches were added next to the lake.<br>• Two gazebos were built on the island so that visitors can better view the lake and animals.<br>• Platforms were built close to the lake.<br>• Plants surrounding the lake were renewed. |
| **Wanshu Hill** | 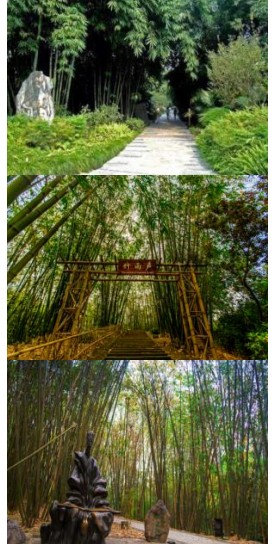 Wanshu Hill is in the southern part of Huanhuaxi Park. Wanshu means "plenty of plants" in Chinese, referring to the various species of trees growing in this area (especially the bamboo). A pavilion lies at the top of the hill. Several statues are distributed on the hill. | 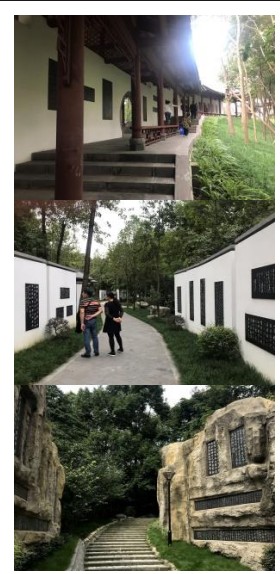 • The old pavilion was replaced with a new one.<br>• Three new cloisters were built and form a 310 m stone gallery. It contains 130 pieces of stone inscribed with poems written by Du Fu.<br>• Hundreds of new stone sculptures inscribed with poems were throughout the park in line with the aims of the Qianshibei project. As a result, many plants on the hill were removed. |
| 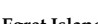**Egret Island** | 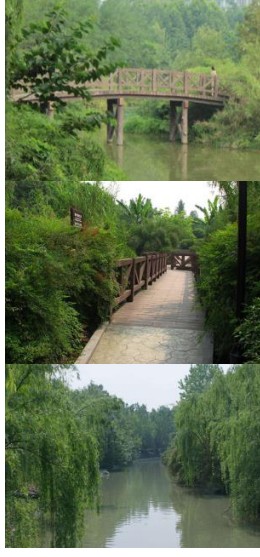 Egret Island lies in the north of Huanhuaxi Park. The island is mainly covered by wetlands, making it an ideal living environment for egrets. The island is divided into several sections, namely the viewing area, feeding area, and isolation area. The viewing area has a large stretch of high forest, and visitors can stroll by the stream to see the egrets. Wooden bridges connect the islands. | 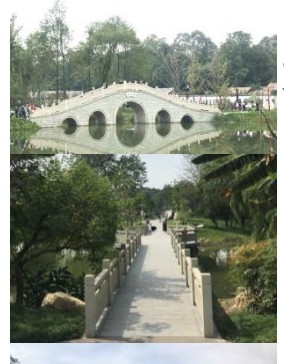 The layout of the island was changed.<br><br>• The wooden bridges were removed and replaced by a new stone bridge, and another stone bridge was built to connect visitors to the Thatched Cottage of Du Fu.<br>• A new trail was built across the water, and all trails on Egret were renovated. Benches were installed along the stream and trails.<br>• Many trees were removed and replaced with new plants.<br>• Stone walls inscribed with poems were built. 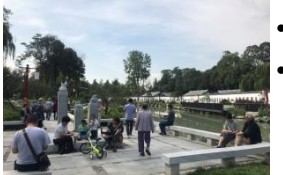 |

### 2.2. Data Collection

As we aimed to perform a participatory mapping of the complete range of CES perceived by park users, we first selected subcategories of CES. Subsequently, data were collected by a field survey in Huanhuaxi Park, including participatory mapping and interviews.

Evaluating CES began with the categories established in the classification of Millennium Ecosystem Assessment, because they are widely used and developed in CES studies [16,39,40]. They are as follows: recreation and ecotourism, aesthetic values, educational values, knowledge systems, spiritual and religious values, inspiration, cultural heritage values, social relations, and sense of place (Table 2). It is notable that knowledge systems and educational values are combined into one category: "educational values". The terms are difficult to differentiate, and this makes them easier to understand. In addition, we further developed indicators and translated them into questions, instead of asking respondents about the service itself because lacking information limits people's ability to evaluate CES [5,41]. For example, we asked "Where are you touched by the beauty of the park?" [42] to indicate aesthetic values (Tables 2 and A1).

Sixty-eight participants participated in this study during September and October 2018. Only participants who visited the park regularly were selected. We defined regular users as people who visited the park at least 1 to 3 times per month, to ensure respondents had adequate experience of the park before and after the renovation in order to improve the accuracy of the assessment. This reduced the barriers of different perceptions caused by different familiarity levels [43]. Participators were selected randomly but with a fairly balanced ratio between women and men, younger and older, etc. Each interview was conducted by the same person and took, on average, 30 to 60 min. The survey contained three parts: (1) a brief introduction of the CES and the purpose of the study. (2) We asked respondents to indicate places where they perceived each CES before and after the renovation on two maps (A3 format, at 1:5000 scale). The paper maps were made based on the field trip, a master plan of the park, a Google satellite map, and a Google historical map, which showed how the park had changed over time. We set 2015 (the year before the renovation) and 2018 (the year when the renovation was finished and the study was performed) to represent the historical and current situation, and to enhance the comparative effects. Subsequently, respondents were familiarized with the maps of the study area and informed about the mapping process. They were then asked to mark the maps with the marker dots denoting a certain service to a site. Each service mentioned in Table 2 was marked in a different color. Participants were allowed to assign up to six marker dots for each service on anywhere they perceived the CES in the case area [41]. (3) Subsequently, we asked what park renovations influenced their perception changes. (4) Finally, we asked a set of questions on the socio-demographic characteristics of the respondents, including age, gender, education, income, and employment (Tables A2 and A3).

The interviews were conducted at different times and with different weather conditions to keep any bias as minimal as possible [44,45]. To maximize response rates, we adopted a series of incentives [41], by giving a small gift to respondents.

### 2.3. Data Analysis

ArcGIS was used for the spatial analyses and Excel was used for the statistical analyses. Specifically, the socio-demographic characteristics of the respondents were classified in absolute numbers and in relative proportions. We also listed the absolute and relative number of the named cultural service dots of all the respondents ($n$ = 68). The maps were scanned, and the location of mapped sticker dots were digitized to enable a spatial analysis in ArcGIS. We displayed the perceived cultural services by respondents on separate maps. "Heat" maps of the spatial concentration of assigned marker dots were generated and calculated by using Kernel Density with Spatial Analyst in ArcGIS. The framework of this study is shown in Figure 2.

**Table 2.** Selected cultural ecosystem services and their definitions [1], and corresponding mapping questions as used in this study.

| Cultural Ecosystem Services | Definition | Mapping Questions |
|---|---|---|
| **Recreation and ecotourism** | Characteristics of living systems that enable activities promoting health, recuperation or enjoyment through active or immersive interactions. | Where do you perform activities (i.e., do sports, walking) and relaxing? [42,46] |
| **Aesthetic values** | Characteristics of living systems that enable aesthetic experience. | Where were you touched by the beauty of the park? [42] |
| **Educational values** | Characteristics of living systems that enable education, training, scientific investigation or the creation of traditional ecological knowledge. | Where do you feel that you learn a lot? [42] |
| **Religious and spiritual values** | Elements of living systems that have spiritual, sacred or religious meaning. | Where do you gain experience about sacred or religious elements? [17,42,47] |
| **Cultural heritage values** | Characteristics or features of living systems that have an existence value, historical value, or that are resonant in terms of culture or heritage. | Where do you feel the historical culture? [48] |
| **Inspiration** | Ecosystems provide rich sources of inspiration for art, folklore, national symbols, architecture, and advertising. | Where inspires you? [17] |
| **Social relations** | Ecosystems influence the types of social relations that are established in particular cultures. | Where do you feel a strengthened bond with others? [42] |
| **Sense of place** | The collection that people feel with recognized features of the environment, including aspects of ecosystem. | Where do you feel a sense of belonging or have a lot of memorable experiences? [46,49] |

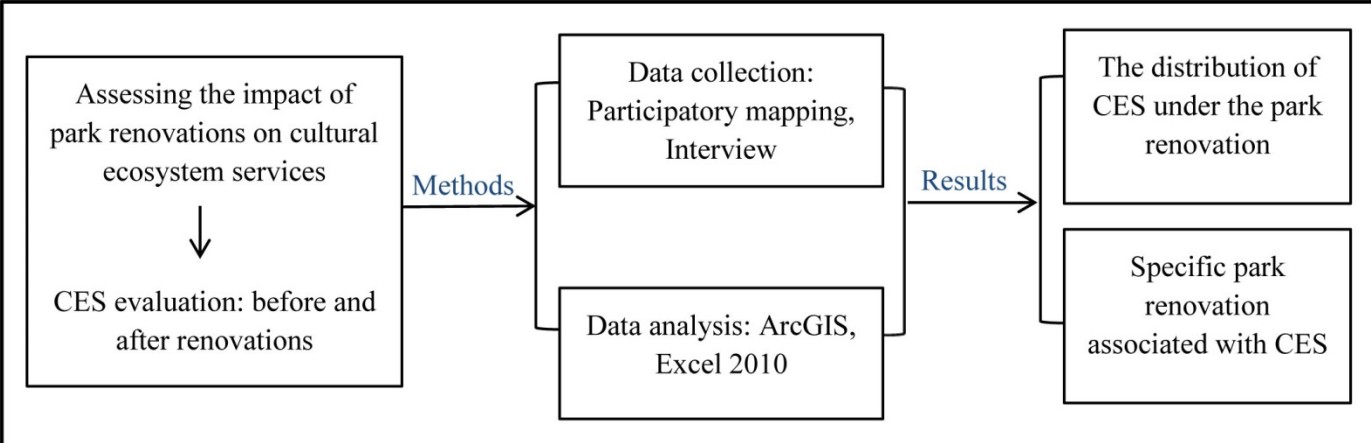

**Figure 2.** The framework of this study.

## 3. Results

In this section, we first present the comparative distribution of CES before and after the park renovation in Section 3.1. Then, Section 3.2 shows what specific park renovations influence the CES.

### 3.1. The Distribution of CES under the Park Renovation

The heat maps (Figure 3) show that CES are mainly distributed in several parts both before and after the renovation, with a higher concentration in the central part, especially the Canglang Lake, followed by the south and north parts, represented by Wanshu Hill and Egret Island. The hot spots were concentrated in Canglang Lake, Wanshu Hill, and Egret Island after the renovation, which correspond to the main renovation locations, while

the distribution of each service changed. Generally, Canglang Lake gained most attention after the renovation. More recreation and ecotourism, aesthetic values, and social relations were found in this area, however, religious and spiritual value, as well as sense of place, decreased. In addition, cultural heritage values and inspiration slightly changed. Wanshu Hill gained more recreation and ecotourism, educational values, religious and spiritual values, cultural heritage values, and social relations, while, aesthetics values, inspiration, and sense of place decreased. Egret Island gained more recreation and ecotourism, cultural heritage, social relations, and sense of place, while religious and spiritual values, aesthetic values, inspiration, and sense of place declined.

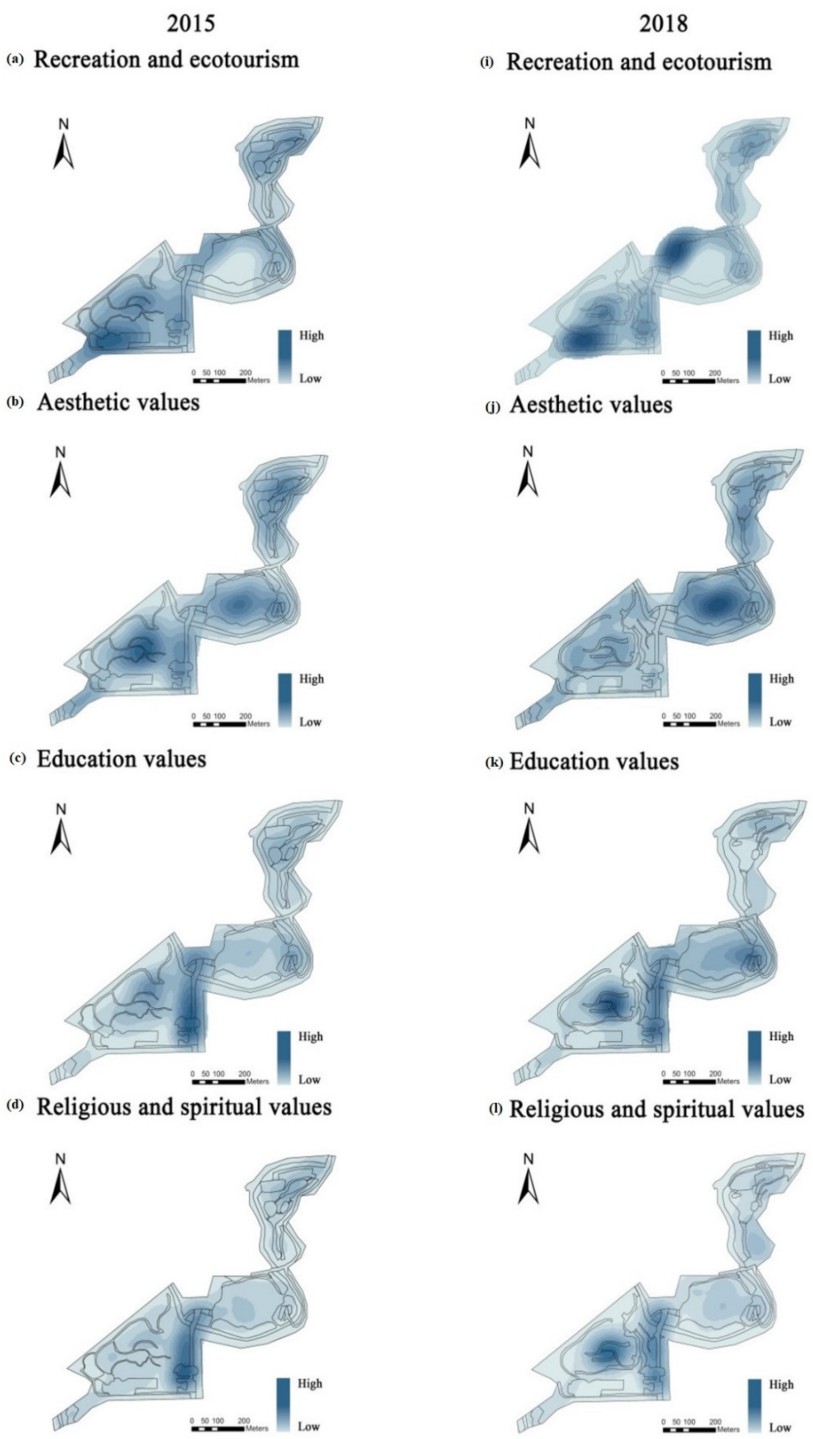

**Figure 3.** *Cont.*

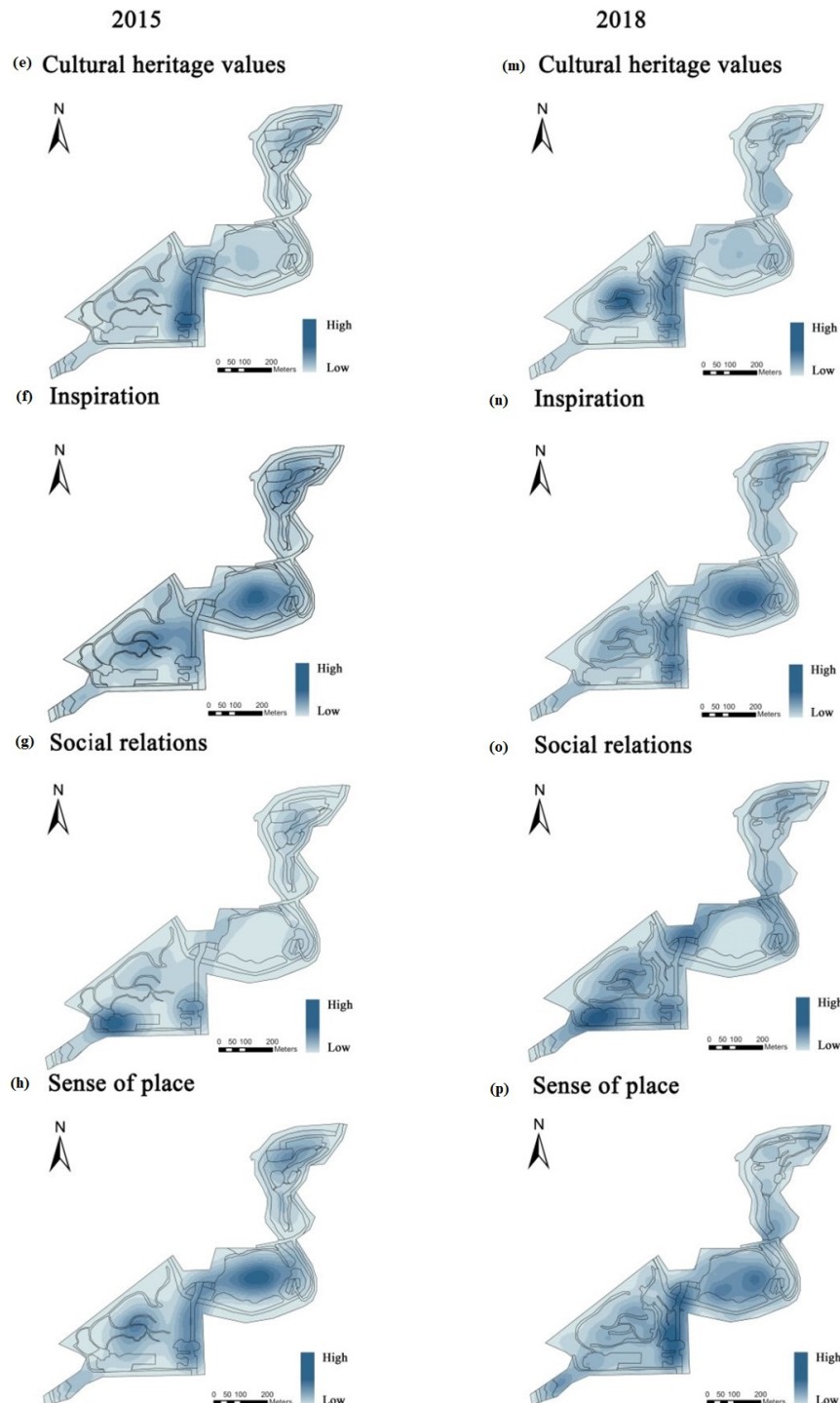

**Figure 3.** The distribution of each CES in 2015 (**a–h**) and 2018 (**i–p**).

### 3.2. Specific Park Renovations Associated with CES

As described in Section 3.1, the distribution of CES changed after the renovation. Here, we present what specific park renovations influenced people's perception based on the content of the interviews.

Figure 4 shows the park features that influenced the interviewees' perceived changes to services after renovations. The lake, architectures (pavilion, gazebo, bamboo pavilion, cloister, etc.), and sculptures (art, murals, statues, etc.) were most commonly mentioned park features that influenced their perception changes. For example, the improvement of the lake was most frequently mentioned as a positive change for providing recreation and

ecotourism, aesthetic values, as well as social relations, along with other improvements such as the clearer water body, more benches, the new platform along the lake, and the diversity of plants. One female adult respondent noted that "the lake is much more beautiful than before, the water is clearer, and there are more places and spaces to sit, thus giving me an opportunity to stay longer and enjoy the scenery and hold conversations with families and friends. I also find myself more willing to talk to strangers who sit near me which I never did before, because I am relaxed in this peaceful, friendly, and safe environment. That's why I marked the dot on the renovation map instead of the map before." Other renovations, such as the new trail and square were often mentioned as having a positive influence on enhancing recreation and social relations. The new bridge had a positive influence on the increase of the aesthetic values and sense of place. The platform and gazebo had a positive effect on the interviewees' sense of inspiration. For example, an amateur photographer stated that he likes taking photos of the lake and birds ever since the park renovation because there are many new kinds of birds to be found surrounding Canglang Lake. Here, his inspiration was related to the renovated lake as well as the gazebo, which served as a bird-watching location. Other features, including infrastructure (toilets, lighting, parking, bins, signs, and bulletins, etc.), the lawn, and play/fitness facilities received the least attention. Besides, not all renovated park features had a positive effect on each service. Figure A1 shows the features that had a negative effect on services. Sculptures, play/fitness facilities, and plants received the most controversial responses. For example, the changes in Wanshu Hill with "too many sculptures and architectures" were most frequently mentioned as having negative effects on aesthetic values, inspiration, and sense of place. In addition, respondents with different socio-demographic backgrounds such as age or gender showed a different preference regarding the renovations. For example, elderly people were more satisfied with the additional pavilions and cloisters compared to younger people. Children and middle-aged women were more satisfied with the added square because it provided space for social activity, which increases social relations and recreational values. People revealed different perceptions of synergies and trade-offs. For example, most young people perceived the recreational and aesthetic values of Canglang Lake as existing in synergy, while some elderly people perceived them as trade-offs.

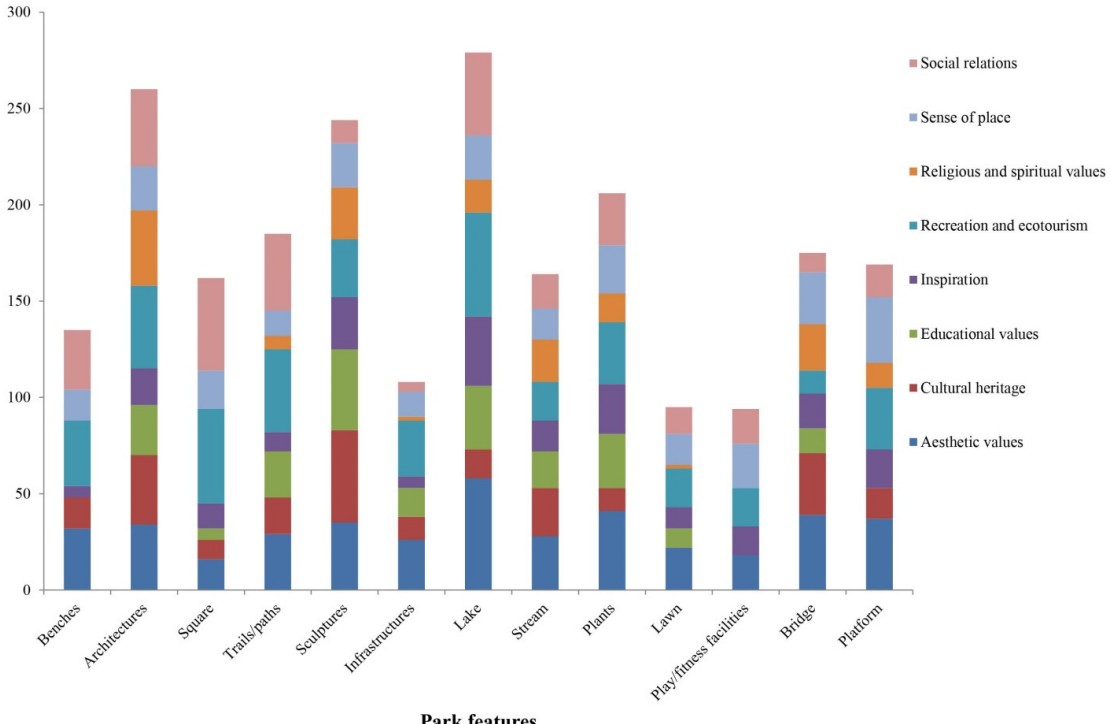

**Figure 4.** Counts of park features mentioned by interviewees that have effects on each service.

## 4. Discussion

It is well known that CES provided by urban parks are important to people. However, how CES are influenced by park renovations is unknown, and how to integrate CES evaluation to support park practice is still a challenge. In this section, we first discuss how CES are influenced by park renovations and explore the potential of integrating CES evaluation to support the park practice. Then, we discuss the method.

### 4.1. Park Renovation Influence on the CES Distribution

This study shows that CES were identified and were distributed unevenly, and different CES were centralized in different locations. CES distribution changed a lot after the renovation. Firstly, the changes corresponded to the main renovated locations. For example, Canglang Lake produced high educational values after the renovation. Wanshu Hill provided more cultural heritage values, religious and spiritual values, as well as social relations. Secondly, different CES were influenced by different park renovations, and one service could be affected by one or more park features, based on individuals' experience and preference. The results show they had many similar perceptions based on their statements. For example, the highly-rated seating areas by Canglang Lake increased their perception of recreational values and social relations. Thirdly, not all the changes had positive impacts on CES: the increase of some CES had a positive or negative effect on other CES. These synergies and trade-offs confirm that the park produces many valuable services that are interlinked and inseparable [5,50,51]. This study further found that the relationships between CES are much more complex than often described. An individual service had relationships with more than one service, and not all services have an equal influence on other services. For example, in Canglang Lake, aesthetic values have a clear and significant influence on recreation and ecotourism, while have less influence on spiritual and religious values. Complex relationships existing among services make it difficult to make decisions regarding park practice. For example, the added recreational facilities in the park can increase the recreational values, while often reducing the natural aesthetic values. In order to use CES as an evidence base to guide park design and management, future studies should address the way CES are associated with each other. The studies on the influence of the specific renovations on CES interactions are encouraged in order to reduce the trade-offs and increase the synergies by managing or adjusting the specific park renovations. The participation of people is encouraged, which allows them to state the complex relationships directly, and figures out the specific features that influence the interactions. It is also notable that synergies and trade-offs not only exist between CES, but also between CES and other ecosystem services (i.e., provisioning, supporting, and regulating services), and their interrelations are also complex and often non-linear [10,52–55]. For example, in Canglang Lake, purified water (a regulating service) and the designed aquatic habitat for plant and animal species (a supporting service) had positive effects on the site's aesthetic values and social relations (cultural ecosystem services). Landscape architects increasingly face new challenges with today's rapid process of urbanization, such as designing resilient landscapes for a changing climate, controlling the impact of natural disasters, and creating a sense of place [56,57]. Meanwhile, clients and the general public are increasingly concerned with ecological functions and environmental conservation [58]. To achieve these multiple and often competing objectives, the integration of the ES framework has the potential to maximize benefits. Although this study placed emphasis on the trade-offs within CES to draw attention to the interactions among CESs, we also suggest that future studies should take all of the services into consideration. This will mean that comprehensive and accurate evaluation outcomes can be attained to better support further practices.

### 4.2. Integrating CES Evaluation to Support Park Design and Management

Integrating CES evaluation into park renovation seeks to integrate considerations of cultural values into park renovation in order to increase the chances of the success of renovation efforts, and further support park practice. Some studies have been conducted

on integrating CES evaluations into land-use planning, urban planning, and urban green infrastructure planning as stated in the Introduction section. However, few efforts have focused on integrating CES in landscape architecture design. Besides, CES have been overlooked for a long time in urban green space projects, which often impedes the achievement of a project's goals [25]. Some project proposals are discussed by a jury, which often consists of local politicians or citizens who can comment on the design proposals and share their thoughts. These inputs are then used to adapt drafts. However, this approach often focuses on investigating general satisfaction with a park, or focusing on one or two CES (e.g., recreation and aesthetic values), which often leads to inaccurate or even false outcomes. A well-designed and acceptable framework for assessing CES or ES is essential for improving this situation. The most common issue is that the majority of projects are authority-oriented and defined by designers or planners, and this often ignores local knowledge, values, and attitudes, claiming that these are subjective aspects. As CES highly depend on people's perceptions and preferences, it is crucial to take these aspects into consideration. The aim of Qianshibei Project was to increase the value of cultural heritage and educational values, but many visitors complained that the added sculptures were too much and that their effects were insignificant, instead decreasing the park's aesthetic values. Hence, full communication with users is important for designers and authorities. The integration of ES or CES evaluations is a useful tool for communicating with users and stakeholders, and shows the possible benefits and how these might change according to different design or management strategies.

This study provides a basis for the incorporation of CES evaluation into urban park design and management in order to maximize CES in cities. The ongoing maintenance of urban parks is an important factor in the effective supply of CES in urban areas. However, the management of urban parks often requires complex and diverse knowledge, which often cannot be easily obtained [59]. Understanding how specific renovations cause changes in CES provides evidence for daily park management. This information provides an opportunity for park managers to maintain the synergies and alter the trade-offs. This is achieved by focusing on the park's features and promoting CES more effectively by managing and adjusting these features. In this case, for example, reducing the number of sculptures in Wanshu Hill may increase the aesthetic value, inspiration, and sense of place, as stated by park users who had a strong perception that too many sculptures destroy the beauty and sense of place. Hence, the relationships between CES and associated park features are crucial in order to achieve successful management. CES can guide landscape architects to rethink their daily designs of urban parks or other urban green spaces.

Although evaluation of the CES after construction will be informative, evaluating the design plans, proposals, or different design scenarios before construction will help designers estimate different values, and select or conceive design alternatives. Many of the assumptions used regarding decision-making involving parks are not stated clearly, and are often based on limited or poor scientific evidence [60]. Discussing and evaluating CES allows interaction and communication among designers, users, and local authorities before making decisions, which can result in more justified decisions. For example, allowing park users to evaluate the design proposals before the renovation may reduce the CES trade-offs.

Another difficulty is that people with different socio-demographic backgrounds, such as gender and age, reveal different perceptions (this is shown in Section 3.2). Other studies on the influence of socio-demographic backgrounds on the evaluation of CES confirm this [49,61]. Failure to account for these factors can result in poorly informed decisions and reduce the provision of multiple services [53,62]. To prevent this, future studies should focus more on revealing the different socio-demographic factors that influence perceptions of CES. Furthermore, it is notable that this study focuses on CES without taking other services into considerations. It is important to further integrate CES evaluation into the ES framework which highlights both ecological and social functions of urban parks.

### 4.3. Methodology Considerations

Participatory mapping is regarded as a useful tool to assess CES [63]. It has emerged as a powerful tool that allows individuals to represent themselves spatially, bringing their local knowledge to support environmental practices [64,65]. It highlights people's preferences, and links the services to a specific location and specific features, which can guide environment protection and investments [66–68]. There are diverse methods used to assess CES in urban parks. One is the "expert-based method", which draws upon the knowledge of experts to deal with complexities and uncertainties, especially in data-poor environments [16]. However, there is growing evidence that design biases and prejudices exist in many expert-led information transfer approaches to environmental evaluations [69]. This is especially true in CES evaluations, because CES evaluations highly depend on people's perceptions and preferences [11]. Having dialogues with people is critical to assessing CES in urban contexts due to the high level of park usage by urban dwellers. We believe that experts who are familiar with jargon and specific techniques are helpful at the beginning of CES evaluations, particularly for obtaining information and stating what they find important regarding CES issues. Later, investigations with people through questionnaires, interviews, or focus groups are necessary to reveal people's perceptions and preferences regarding CES. In addition, other methods such as the "social media-based method" (based on photos related to CES that are shared on social media) [66,70,71], or online participatory mapping [8,72], are flexible and less time-consuming. However, this study focuses on the changes to CES influenced by park renovations that are complicated. Face-to-face interviews are crucial for attaining accurate outcomes, and considering that a large number of interviewees were elderly and unfamiliar with newer techniques, a field survey conducted with paper maps were more acceptable to these people than online mapping.

Studies linking CES to specific renovations ground the theoretical CES into practical information for park managers to support their daily maintenance and management practices, such as adjusting renovations as discussed in Section 4.2. We based our participatory mapping approach on historical and current maps instead of doing a follow-up study. The follow-up evaluation is often time-consuming and the evaluation needs to adjust to unpredictable changes that are often out of the researchers' control [73]. Researchers need to be flexible with timetables and funding to cope with the changes. The comparative mapping study after the renovation is flexible and easy to implement by following the structure of the mapping approach, which is less time-consuming.

The comparison of the two heat-maps, and their links to particular park features by interviews, give park managers a clear and whole picture of the changes of CES in order to identify the key locations and problems, and hence make corresponding management strategies or solutions. The problem is that people might be influenced by their memories, which may affect the accuracy of the evaluation outcomes. However, on the other hand, the comparison of two maps enhances people's perception because it gives them a chance to pay attention to their changed surrounding environment, what it means to them, and what are the differences of their perceptions or preferences before and after the renovation. Hence, this method can be adopted in other relevant studies such as retrospective pretest evaluations of urban green spaces. The inclusion of park users in the research process is important for data collection and will greatly assist the dissemination of results to key stakeholders (park users, managers, designers, etc.). This full participation empowers practitioners and stakeholders to effectively communicate which CES are important to them and how they might be affected by design or management options. This helps designers to find sustainable solutions and enhance the adaptive capacity of their designs. Participatory mapping can help target less-valued areas or important park features for redesign and management [74]. This mapping process can help designers better understand the interactions between humans and the environment, thus promoting a willingness to protect the environment. Previous research also has suggested that the participation of stakeholders can improve their awareness of protecting the environment, and the facilitation of knowledge transfer between different stakeholders (researchers, designers, managers, or users)

is critical to success in designing and managing urban parks [59]. This requires carefully identifying the key stakeholders and their needs [15]. Analyzing differences in assessing CES can better understand mutual and disputing interests between stakeholders [25]. This study also reflects the urgent call to address the challenges in a transdisciplinary way because the problems are complex. This includes encouraging more knowledge, methods, and techniques about how to enhance human perceptions and memories in order to improve the evaluation accuracy. For example, focus groups, or deliberative techniques can be introduced in the mapping exercise, focusing on better expressing preferences by providing more time and information to participants for discussions, and to ensure they become more familiar with CES [16].

## 5. Conclusions

This study evaluates the impacts of park renovations on CES in Huanhuaxi Park. The results show that the park renovations do affect the provision and distribution of CES. The results also show evidence that specific park renovations influence the provision of specific CES, but that the influences are complex, showing trade-offs and synergies. To conclude, we first highlight that the results can guide park design and management by adjusting the specific park renovations. Second, we encourage more investigations of the complex relationships among CES, as well as those between CES and other ES, which were influenced by park renovations. Third, we highlight integrating CES evaluation into practice, such as using them to justify the design proposals or evaluating alternative plans before the construction. Moreover, we highlight combining participatory mapping and interviews to assess CES, which can link the CES to a specific site and specific renovations. We also encourage a comparative study by using historical and current maps to support retrospective pretest evaluations. We also suggest exploring more methods and techniques such as focus groups, or deliberative techniques to enhance the comparison effects.

**Author Contributions:** X.C.: Conceptualization, Methodology, Investigation, Formal analysis, Writing—Original draft preparation; S.V.D.: Methodology, Writing—Reviewing and Editing; P.U.: Writing—Reviewing and Editing. All authors have read and agreed to the published version of the manuscript.

**Funding:** This research received no external funding.

**Informed Consent Statement:** Informed consent was obtained from all subjects involved in the study.

**Data Availability Statement:** The data presented in this study are contained within this article.

**Acknowledgments:** The authors would like to acknowledge all the participants who joined this study.

**Conflicts of Interest:** We declare that we have no conflict of interest to this work. We have no financial and personal relationships with other people or organizations that can inappropriately influence our work.

## Appendix A. Supplementary Data

**Table A1.** Protocol of mapping.

| Cultural Ecosystem Services | Questions |
| --- | --- |
| Recreation and ecotourism | Where in this park do you perform activities (i.e., do sports, walking) and relaxing before the renovation? And after the renovation? Please state reasons? |
| Aesthetic values | Where in this park were you touched by the beauty before the renovation? And after the renovation? Please state reasons? |
| Education values | Where in this park do you feel that you learned a lot before the renovation? And after the renovation? Please state reasons? |
| Religious and spiritual values | Where in this park do you gain experience about sacred or religious elements before the renovation? And after the renovation? Please state reasons? |
| Cultural heritage values | Where in this park do you feel the historical culture before the renovation? And after the renovation? Please state reasons? |

**Table A1.** *Cont.*

| Cultural Ecosystem Services | Questions |
|---|---|
| Inspiration | Where in this park inspires you before the renovation? And after the renovation? Please state reasons? |
| Social relations | Where in this park do you feel a strengthened bond with others before the renovation? And after the renovation? Please state reasons? |
| Sense of place | Where in this park do you feel a sense of belonging or have a lot of memorable experiences before the renovation? And after the renovation? Please state reasons? |

**Table A2.** Socio-demographic backgrounds of participants (as the respondents are Chinese, we translated the questionnaire to Chinese, and then translated the results into English for the convenience of readers of this journal).

| Questions | Response |
|---|---|
| You are | Female |
| | Male |
| Your age is | 12–18 |
| | 19–29 |
| | 30–39 |
| | 40–64 |
| | ≥65 |
| What is your highest educational qualification? | <Bachelor's degree |
| | Bachelor degree |
| | Master degree |
| | ≥PhD |
| What is your occupation? | Employed |
| | Retired |
| | Housewife/-husband |
| | Student |
| | Unemployed |
| | Others |
| What is your income? | None |
| | 0–2999 RMB |
| | 3000–4999 RMB |
| | 5000–9999 RMB |
| | ≥10,000 RMB |
| How often do you visit green space in summer time? | Everyday |
| | 4–6 times/week |
| | 2–3 times/week |
| | Once a week |
| | 1–3 times/month |
| How often do you visit green space in winter time? | Everyday |
| | 4–6 times/week |
| | 2–3 times/week |
| | Once a week |
| | 1–3 times/month |

**Table A3.** Socio-demographic characteristics of the 68 participants.

| Category | Responses | Interviewees (*n*) | Percentage |
|---|---|---|---|
| **Gender** | Female | 31 | 46% |
| | Male | 37 | 54% |
| **Age** | 12–18 | 6 | 9% |
| | 19–29 | 9 | 13% |
| | 30–39 | 24 | 35% |
| | 40–64 | 19 | 28% |
| | ≥65 | 10 | 15% |
| **Education** | <Bachelor's degree | 37 | 54% |
| | Bachelor degree | 20 | 30% |
| | Master degree | 9 | 13% |
| | ≥PhD | 2 | 3% |
| **Occupation** | Employed | 27 | 40% |
| | Retired | 11 | 16% |
| | Housewife/-husband | 8 | 12% |
| | Student | 9 | 13% |
| | Unemployed | 2 | 3% |
| | Others | 11 | 16% |
| **Income** | None | 18 | 27% |
| | 0–2999 RMB | 9 | 13% |
| | 3000–4999 RMB | 17 | 25% |
| | 5000–9999 RMB | 19 | 30% |
| | ≥10,000 RMB | 5 | 7% |
| **Frequency visiting: summer time** | Everyday | 6 | 9% |
| | 4–6 times/week | 13 | 19% |
| | 2–3 times/week | 7 | 10% |
| | Once a week | 14 | 21% |
| | 1–3 times/month | 28 | 41% |
| **Frequency visiting: Winter time** | Everyday | 2 | 3% |
| | 4–6 times/week; | 11 | 16% |
| | 2–3 times/week | 8 | 12% |
| | Once a week | 13 | 19% |
| | 1–3 times/month | 34 | 50% |

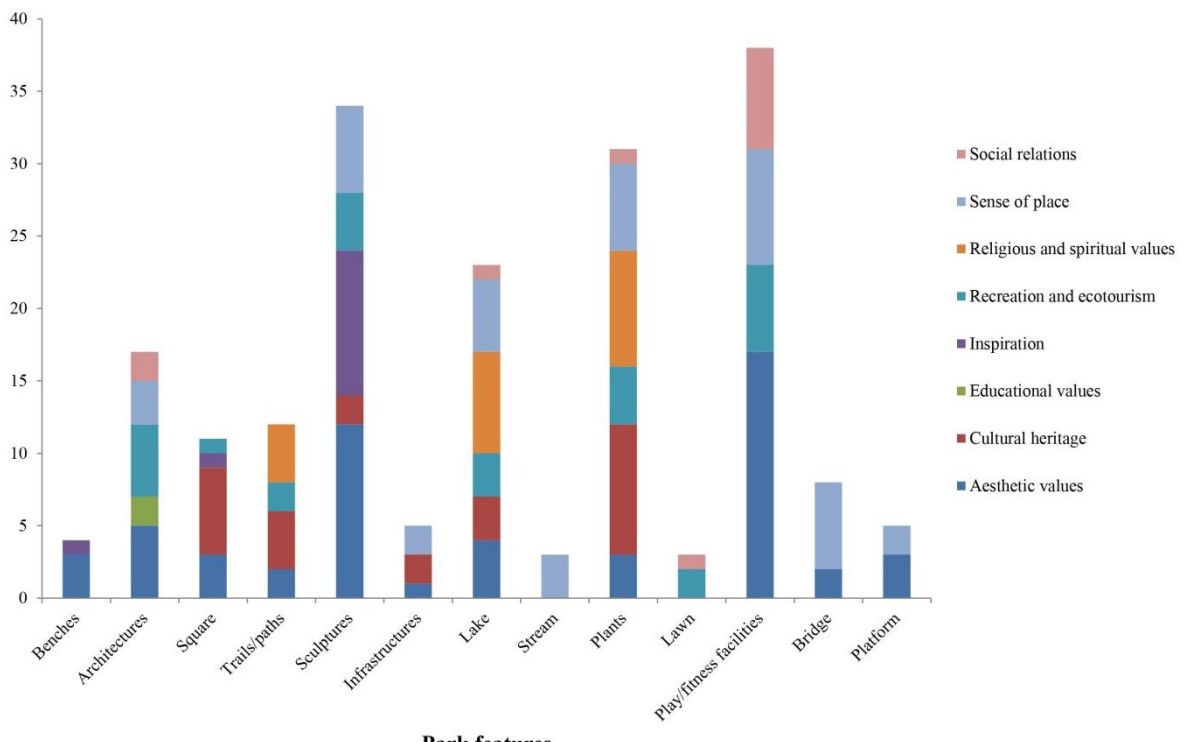

**Figure A1.** Counts of park features mentioned by interviewees that have negative effects on each service.

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
