# Peer review of "Assessing the Impact of Park Renovations on Cultural Ecosystem Services"

_land, doi:10.3390/land11050610_

Round 1

Reviewer 1 Report

This paper reflects on the impact of urban green space on the provision and distribution of cultural ecosystem services. The paper is well written, and addresses important dilemmas on synergies and trade-offs among complex cultural ecosystem services. The supplementary material is clear and supports the text. The concluding remarks are well developed and highlighted. There are only three minor issues to be addressed:

- references to the definition of cultural ecosystem services should be added in the caption or in the Table 2,

- the sentence in lines No. 290–292: it is not clear what the term »they« refers (For example, in Canglang Lake, aesthetic values are recognized has a clear and significant influence on recreation and ecotourism,  while they have less influence on spiritual and religious values.)

- although the text is well written I recommend authors to read it again and check it because in some (rare) places the sentences are strangly constructed  (for example …aesthetic values are recognized has a clear and significant…)

I wish the authors successful publishing the paper!

Reviewer 2 Report

The topic is interesting from the point of view of of park renovations and cultural services.

The study has good potential but the manuscript needs some structural corrections.

Specific comments:

  1. It would be important to include the issue related to the ecological importance of parks. In the 21st century, it is unacceptable to plan or change the functions of parks without taking into account ecological aspects, but only introducing social services. Therefore, it would be important to assess whether the park has increased ecological and health-related values, and not only aesthetic or cultural issues, after the park's revitalization.
  2. The introduction lacks information about the global significance of the presented research results. I mean, how the international audience could benefit from the solutions presented by you.
  3. I would recommend adding a framework or flow chart to the Methods section to explain the research stages.
  4. The discussion definitely lacks a comparison to other studies in this topic and indications of references.
